# Cultural Tourism in a Post-COVID-19 Scenario: The French Way of Saint James in Spain from the Perspective of Promotional Communication

**Clide Rodríguez-Vázquez** [1,*] **, Pablo Castellanos-García** [2] **and Valentín Alejandro Martínez-Fernández** [3]

1    Faculty of Economics and Business, University of A Coruña, Campus de Elviña, s/n, 15071 A Coruña, Spain
2    Department of Applied Economics, University of A Coruña, 15071 A Coruña, Spain
3    Department of Business, University of A Coruña, 15071 A Coruña, Spain
*    Correspondence: c.rodriguezv@udc.es

**Abstract:** Tourism has been one of the sectors most affected by the COVID-19 pandemic. One of the side effects of the pandemic is the demand for safe and quiet spaces, giving rise to the search for a new lifestyle, "slow living", which could represent an opportunity for cultural tourism. In this context, the main objective of this article is twofold: (i) to establish the relevance of cultural tourism for residents in Spain for the autonomous communities along the French Way of Saint James and (ii) to determine their behaviour on their institutional tourism promotion websites. For our analysis, we use equality of means tests and ANOVA (for data from 2002–2020), as well as measures of positioning, engagement, origin of the audience and access devices (for data from 2020–2021). The main conclusion is that the Way of St. James does not act as a driving force for cultural tourism, even though the websites of tourism promotion organisations have experienced a remarkable growth in their use. This article develops an original relation of cultural tourism through an analysis of the French Way of St. James in Spain and the web positioning of official tourism promotion organisations before and during COVID-19.

**Keywords:** promotional communication; cultural tourism; French Way of Saint James; French Way; COVID-19

## 1. Introduction

Over the last few decades, world tourism demand has experienced a significant growth from 280 million in 1980 to 1.5 billion in 2019 [1]. This was, among other factors, due to the population explosion, the increase in world income, the improvements in communications and infrastructure, technological advances, new business models, more affordable travel costs and the simplification of the visa regime, which have favoured the development of new consumption needs linked to tourism.

Economic growth has traditionally been linked to the secondary or industrial sector, but more and more countries have adopted models where the third sector in general and, within this, tourism in particular, is the engine of their economy. A large part of their GDP is tied to the arrival of visitors and their spending. An example of this is Spain, a world power in terms of tourism. This country had 83.5 million international tourists in 2019 [2], 1.1% more than the previous year, with an expenditure of 92,278 million euros [3], which means an increase of 2.8%. These figures represent a record number of visitors and tourist expenditure, placing Spain as the second country in the world in terms of international tourist arrivals, after France [4].

Only the pandemic put the brakes on this exponential growth. In 2020, due to the restrictions resulting from the global emergency generated by COVID-19, the contribution of tourism to Spanish GDP fell from 12.4% in 2019 to 5.5%. The number of tourists visiting Spain this year was 18.9 million, which was 64.5 million less than in 2019 and implies a

decrease of 77.3% [2]. After a year of the pandemic and a slowdown in the sector where it has become clear that health quality is a determinant of international tourism revenues [5], the crisis has overshadowed the global market and the growth expectations.

Faced with the bleak figures for tourist movements in 2021, the WTO, knowing that many millions of people and businesses depend on this sector, has required on several occasions from the international community to take strong and urgent action to ensure a more promising 2021 [6]. As travel resumes in some parts of the world, the limited connectivity and low consumer confidence, the uncertainty about the evolution of the pandemic and the impact of economic downturn pose unprecedented challenges to the tourism sector, leading to increased competition between destinations.

Since in Spain tourism is a key sector of its economy, it needs to redefine itself in terms of its outlook and its strategies due to the internal and international crisis. In this context, cultural tourism, as an alternative to sun and beach tourism, can be a tool for competitiveness and economic growth, with great potential for job creation and an opportunity to regain market share, at both the national and international level.

Thus, being aware of the difficult situation of the tourism sector, we have verified that no research has empirically examined how cultural tourism can function as a locomotive of tourist demand in a crisis situation through the example of the French Way of St. James in Spain. The question we are investigating is whether this type of tourism works as a key motivation for doing the French Way of St. James, in addition to seeing how public tourism promotion institutions have developed their work of approaching tourists to provide them with all the necessary information to know the route.

This contribution intends, as its main objective, to establish the relevance of cultural tourism as a key travel motivation for Spanish residents in the CCAAs through which the French Way of St. James runs, and how they behave on the institutional websites of tourism promotion agencies. As secondary objectives, it pursues:

- To ascertain the degree of relevance of tourist trips made by residents in Spain, mainly for cultural reasons, to the communities through which the French Way of St. James runs, both inland (Navarra, Aragón, La Rioja and Castilla y León) and coastal (Galicia).
- To determine the web positioning and online attraction potential of the ACs through which the French Way of St. James passes. The aim is to contribute to developing a diagnosis of a certain reality of the cultural tourism sector in Spain in line with the importance of the online channel and to establish which communities have better web positioning and exert greater power of attraction.

## 2. Literature Review

### 2.1. Cultural Tourism

Cultural tourism is a type of tourism that obeys the motives and needs of the consumers themselves, so defining its boundaries is highly subjective [7]. There is no consensus in literature on the limits of this concept [8]. Since the middle of the last century, authors such as [9–11], due to the complexity of the relationships between the components involved in this term, analysed it from different perspectives and to do so, they established common elements that make up its definition.

This type of tourism includes two areas, tourism and culture, which have gone through different phases of evolution, conceptual and theoretical encounters and misunderstandings until they formed the categories of tourism/cultural tourism [12].

From this integrative perspective, many authors have focused on the importance of cultural tourism in different aspects: Richards [13] highlighted the offer of this type of tourism as a possible tool for heritage conservation; Chevrier and Clair-Saillant, ref. [14] and Herrero [15] spoke of its two-dimensionality through the market and consumption and Camarero and Garrido [16] emphasised the leading role given to the visitor in the generation of innovative models of visits where the creation of experiences is an important asset. Years later, the National and Integral Plan for Spanish Tourism (2012–2015) [17]

pointed out the importance of cultural resources as an offer for Spain and alluded to strategic lines on experiential and creative approaches and product management.

Heritage protection determined the origin of the first [18], but another series of factors influenced the conceptualisation of this category of tourism, such as the opening up of new ways of life in a context of disruptive changes in the economic, social and cultural spheres as economic resources and the levels of education and culture increased. This led to new forms of travel, for example: fragmenting holidays; short breaks or short-stay trips; more active tourism, as well as a new taste for knowledge during travel time; the liberalisation of air transport; and the improvement of road networks and airport infrastructures for greater connectivity.

In 1995, the WTO [19] defined cultural tourism as all movements of people to satisfy the human need for diversity, aimed at raising the cultural level of the individual, facilitating new knowledge, experiences and encounters.

The ICOMOS charter on cultural tourism [20] again gives a broader view of the concept by insisting on cultural exchange and present-day societies as a form of tourism: "Domestic and international tourism continues to be among the foremost vehicles for cultural exchange, providing a personal experience, not only of that which has survived from the past, but of the contemporary life and society of others".

In December 2018, to demonstrate the importance of cultural tourism nowadays, the third conference on this category of tourism, jointly organised by the WTO and the UNESCO, was held in Istanbul; it highlighted 'the support for culture as a driver for safeguarding living heritage, catalysing creativity in cities and extending the socio-economic benefits of tourism to all cities'. The Committee on Tourism and Competitiveness (CTC), at the 22nd Session of the WTO General Assembly, adopted as recommendations some operational definitions used in the tourism value chain, as well as of various tourism typologies in order to contribute to the establishment of a common basis for a harmonised understanding. According to the definition adopted by the UNWTO General Assembly at its 22nd session, cultural tourism entails:

> *A type of tourism activity in which the essential motivation of the visitor is to learn, discover, experience, and consume the cultural attractions/products, both tangible and intangible, of a tourist destination. These attractions/products refer to a set of material, intellectual, spiritual and emotional elements distinctive of a society encompassing arts and architecture, historical and cultural heritage, gastronomic heritage, literature, music, creative industries and living cultures with their ways of life, value systems, beliefs and traditions. [21]*

### 2.2. Changes in Tourist Consumer Trends Post COVID-19: More Sustainable and Sensitive Tourists

Tourism has been one of the sectors hit hardest by the COVID-19 pandemic and it is uncertain when this situation will end, or whether it will be possible to return to the so-called new normality.

In an interview with Bachiller [22], José María López Morales, Vice-Dean of the Faculty of Economics, Business and Tourism at the University of Alcalá, attributed to the great resilience of tourism the fact that the industry adapts itself and starts generating new ideas and proposals that result in a strong resurgence in the flow of tourists and visitors. It is time to think about how to reinvent tourism, to live through the situation strategically and to keep tourist activity with the least possible losses so that the sector can survive.

In the short term, and when the new normality arrives, there will be changes in tourists' behaviour, which have already begun to emerge during the period of confinement and, at present, this can become an opportunity for the tourist sector. Although, as stated by Arbulú, Razumova, Rey-Maquieira and Francesc Sastre [23]: "We must lso consider the demand shock caused by the drop in consumption as a result of the drop in income and lower export demand".

Beyond the changes stipulated by the regulations created in this respect on mobility, safety distance or use of vaccine passports that reduce perceived health risk and influence decision making [24], tourists prefer environments where they can feel safe, calm and where they can be healed in body and soul during turbulent periods, with a preference for rural areas and familiar environments to make up for the insecurity caused by the uncertainty surrounding the health situation.

A new model of tourism linked to sustainability and safety may characterise this health crisis and will have a different impact on inland regions and coastal regions, where many more foreign tourists travel, because air travel is restricted; however, it may also affect large cities if people decide to avoid crowds.

A side effect of COVID-19 has been a cultural one, forcing us to slow down our pace of life considerably and to stay in safe spaces. In this new situation of forced inactivity many people have begun to adopt a new lifestyle, the so-called slow living or slow life [25]. This is a new trend among consumers that began in 1986 with the slow food movement in Italy, formerly called Arcigola, and which grew up to the point where it now has more than 100,000 members in 132 countries. This initiative has been joined by other elements that have made slowness a revalued characteristic in many areas of life for thousands of people. This trend has grown exponentially in the online sphere but can be adapted to new travellers as it is closely related to escapism, safety, contact with nature, a simple life, minimalism and a sense of fulfilment.

At the most basic level, tourist destinations linked to this trend through their offer, as is the case of those integrated in the French Way of St. James (i.e., the main pilgrimage route of Europe, coming from France up to the cathedral of Santiago de Compostela, in the NW of Spain), can take advantage of it by adjusting their insights to connect with consumers looking for trips with these proclamations.

This model is likely to change the status quo, transforming the current situation into a medium-term opportunity as tourists decide to opt for activities related to a healthier, more spiritual, or more ecological tourism, as is the case with organic wine tourism [26] or the heroic viticulture that we find on the French Way of St. James as it crosses Galicia. The most important thing is to realise that there has been a change in consumers' behaviour in general and in cultural tourism consumers, reflecting new desires and needs. This trend provides insight into the current mood of consumers, which can help destination management professionals know how open their audiences are to new messages and act more strategically to achieve their objectives. Destinations can use these insights to readjust and adapt their offer.

Perhaps this is the time for the managing bodies of the destinations along this ancient route to promote and invest in projects to reach potential tourists, to awaken an illusion that is more necessary than ever and to provoke the desire to follow the route now that it is once again possible.

*2.3. The French Way of St. James: Opportunity through Change in Trends?*

It is within this context where the development of cultural tourism is imbricated with the French Way of St. James in accordance with its declaration as a European Cultural Itinerary established by the Council of Europe in 1987. It is favoured because, for centuries, the pilgrims who made the French Way of St. James returned to their places of origin with a strong cultural identity after discovering different traditions or new lifestyles, languages or heritage, thus turning the routes of the French Way of St. James into a symbol that reflects a model of cultural cooperation for the whole of Europe.

During the twentieth century, the French Way of St. James was subject to an intensive promotional campaign, which broadened its significance by turning it into a tourist product and led to it being declared the 'First European Cultural Route' by the Council of Europe, in 1987, and a UNESCO 'World Heritage Site', in 1993. In 1998 and 2015 it did the same for the Ways of St. James in France, as well as for several other ones in the north of Spain [27].

Against this backdrop, the outlook for the post-COVID-19 tourist environment, we considered it relevant to analyse the situation of the French Way of St. James in the inland (Navarra, Aragón, La Rioja and Castilla y León) and coastal (Galicia) autonomous communities (ACs henceforth) through which it passes in Spain and the role of institutional promotional websites as a support for visibility, interaction with tourists and, ultimately, a channel on which to build a long-term relationship with them.

The choice of the French Way of St. James is because it is the most popular route in the Jacobean scope (seven out of every ten pilgrims). The transit of pilgrims over the centuries has made it a cultural, artistic, and sociological route towards Apostle St. James's tomb, discovered in the 9th century, although the French Way of St. James was laid out, as we know it today, at the end of the 11th century. The French Way of St. James aroused such interest in Medieval Europe that as early as 1135 the Codex Calixtinus became a pioneering travel guide, containing detailed information on this route.

Marchena (1993) [28] defined the Way of St. James as a tourist product because it can be considered one of the first tourist routes in history, as it has all the necessary elements strictu sensu: accommodation (pilgrims' hostels), complementary services (lodgings, inns . . . ), tourist resources (monumental, historical, natural . . . ). Girish and Lee [29] underline the fact that St. James' Way offers pilgrims the opportunity to gain authentic experience, so to maintain the authenticity of the different routes, the expectation of the pilgrims should be met by maintaining the trails, restoring the historical buildings, cathedrals, and churches, and providing other amenities.

On the other hand, the first decades of the 21st century are marked by a global conception of thought and economy, by culture and entertainment, as well as by digital technological development. About the latter trend, tourist managers and promoters must be aware of the important advantages provided by interactive media such as websites when distributing and marketing their tourist resources. Hence, the analysis of the role of promotional websites related to the French Way of St. James seems an issue of great interest.

### 3. Materials & Methods

To achieve the aforementioned objectives, we used a dual methodology: (i) statistical tests of difference of means and analysis of variance and (ii) study of institutional websites by means of the online tool SimilarWeb

The tests of difference of means (*t*-tests), which are one of the most common tests in statistics, are used to determine whether the means of two groups are equal to each other.

Analysis of variance (ANOVA) is a statistical analysis tool that splits an observed aggregate variability found inside a data set into two parts: systematic factors and random factors. The systematic factors have a statistical influence on the given data set, while the random factors do not. The ANOVA is able to determine the influence that independent variables have on the dependent variable.

SimilarWeb ( https://www.similarweb.com, accessed on 1 March 2020) enables the analysis of the traffic and behavior of users on websites and apps. It ranks websites and apps based on traffic and engagement metrics. The ranking system covers 210 categories of websites and apps in 190 countries and was designed to be an estimate of a website's popularity and growth potential.

On the one hand, the most relevant statistical results in the cultural sphere were selected from different statistical sources considering the main indicators for the period 2002–2019 to achieve an objective understanding of the situation of cultural tourism in Spain and its evolution within the Spanish cultural panorama. Data from the Cultural Statistics Yearbook published by the Ministry of Culture [30] were used and analysed by means of statistical techniques, specifically the mean-difference test and one-factor analysis of variance (ANOVA).

This yearbook is structured in two large blocks. The first includes estimates affecting different cultural sectors: employment and companies, public and private financing,

intellectual property, foreign trade, tourism, education, and cultural habits. The second offers more specific information of some of them, such as: heritage, museums, archives, libraries, books, performing arts, music, cinema and video. It also analyses the estimates about tourism and specifically tourist trips made mainly for cultural reasons.

From this yearbook, for our empirical analysis we chose as the variable of interest the trips made by residents in Spain mainly for cultural reasons according to destination, expressed as a percentage of total trips for leisure, recreation or holidays. We did not consider trips for other non-recreational reasons, such as work, where individuals are obviously very restricted in their possibilities to visit monuments, etc.

The data used correspond to the seventeen Spanish ACs. The autonomous cities of Ceuta and Melilla were not included, due to lack of data. Based on the percentages of cultural tourism in each year in the different ACs, statistical tests of difference of means and analysis of variance were applied to test three hypotheses, as indicated below.

Firstly, given the importance of sun and beach tourism in Spain, we tested whether there are significant differences between the coastal and inland regions in terms of the average percentage of cultural tourism in each one.

Next, we checked whether there are appreciable differences between three different groups of ACs: inland regions along the French Way of St. James, coastal regions and inland regions out of this pilgrimage way.

Finally, we analysed, within the group of ACs of the French Way of St. James, whether there are relevant differences between the only one with coastline (Galicia) and the inland ACs (Aragón, Castilla y León, Navarra and La Rioja).

On the other hand, by virtue of the situation experienced after the enactment of the Royal Decree 463/2020 of 14 March [31] due to the pandemic caused by COVID-19, Spanish consumers' behavior and the digital scene have changed rapidly. It is therefore interesting to analyse whether the confinement has led to a closer approach to technology, which would have allowed Spanish tourists to relate more closely to the websites of tourist promotion organisations. This should have prompted these institutions to adapt to these changes and to improve their position, which would be beneficial once the situation returns to normality.

For this second analysis, the online tool SimilarWeb was used. It allows us to know the main data of a given internet site regarding its positioning, audience and type of traffic. In this research, it allows us to obtain measurements of the official webs for tourist promotion of the communities through which the French Way of St. James runs (Table 1). The time interval considered for this study is from 1 March 2020 to 31 May 2020 (3 months of confinement) and from 1 March 2021 to 31 May 2021 (Xacobean Holy Year), which allows for a comparative analysis.

**Table 1.** Web sites of the ACs of the French Way of Saint James in Spain.

| Autonomous Community | Url Site Web |
|---|---|
| Navarra | www.turismonavarra.es |
| Aragón | www.turismodearagon.com |
| La Rioja | www.lariojaturismo.com |
| Castilla y León | www.turismocastillayleon.com |
| Galicia | www.turismo.gal |

The image of the state of community websites results from analysing four parameters and their associated variables [32]. We studied (Figure 1) the positioning of community websites at the international and national level; their engagement or interaction level and commitment degree between the user and the destination/brand through number of visits, duration and bounce rate (tells us what percentage of the visits that enter a website end up leaving it without having visited other pages of that site); the origin (national and international); and finally, the access devices (computer or other mobile gadgets).

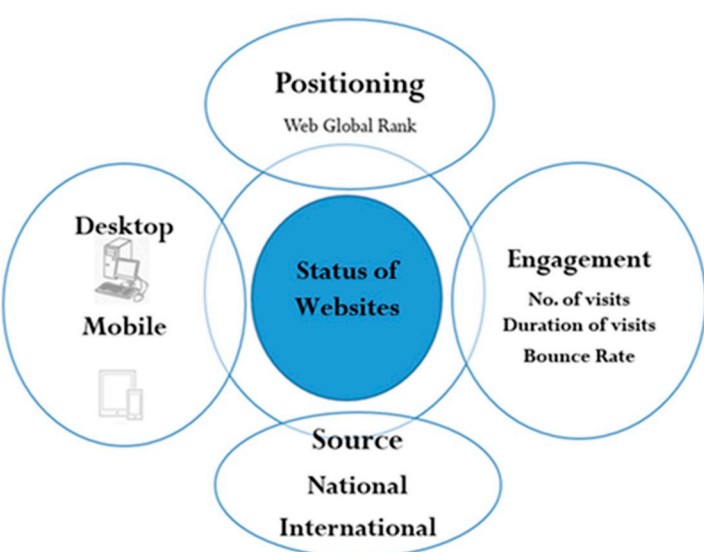

**Figure 1.** Parameters and associated variables of analysis.

## 4. Results

### 4.1. Test of Difference of Means and ANOVA

As mentioned above, the first step is to determine whether there are significant differences between coastal and inland regions in terms of the average percentage of cultural tourism in each region. In view of Table 2, it seems that cultural tourism is higher in inland regions than in coastal regions. However, to corroborate this in a rigorous way, statistical techniques must be applied.

**Table 2.** Descriptive statistics of the coastal and inland ACs.

| Cultural Tourism | N | Average | Standard Deviation | Standard Error | Minimum | Maximum |
|---|---|---|---|---|---|---|
| Coastal | 18 | 11.0086 | 1.2775 | 0.3011 | 8.3300 | 12.5500 |
| Inland | 18 | 13.8957 | 2.4968 | 0.5885 | 10.9000 | 19.8857 |
| Total | 36 | 12.4522 | 2.4421 | 0.4070 | 8.3300 | 19.8857 |

Since there are only two groups, applying ANOVA is unnecessary: a *t*-test for two independent samples can be used. In this test, several options are available, depending on whether the variable analysed is normally distributed, and whether equality of variances can be assumed.

Regarding the first question, the Shapiro–Wilk test for normality found that, at the 5% significance level, the hypothesis of normal distribution could not be rejected for both coastal and inland ACs. As for the second question, Levene's test concluded that, at the 5% level, the hypothesis of the equality of variances in both types of ACs could be rejected. Therefore, we had to apply a test on the difference of means of two independent normal populations whose variances are unknown and different, which led to the conclusion that, at the 5% level, we could claim that cultural tourism is more relevant in the inland ACs. One possible interpretation of this result is that in inland ACs, the lack of the sun and beach tourism option makes individuals consume more cultural tourism, whereas, on the contrary, in coastal ACs, sun and beach tourism turns aside the consumption of cultural tourism.

Next, we studied whether there were appreciable differences between three different groups of ACs: the inland ones crossed by the French Way of St. James and the coastal and inland ones outside this route.

According to the descriptive data (see Table 3), on the one hand, cultural tourism seems to be lower in the coastal ACs through which the French Way of St. James does not

pass; on the other hand, in the inland ACs not crossed by the French Way of St. James, the level of this type of tourism is higher than in the ACs through which the French Way of St. James runs. However, to rigorously verify these assessments, statistical techniques must be applied.

**Table 3.** Descriptive statistics of the ACs of the French Way and the coastal and inland regions through which the French Way does not pass.

| Cultural Tourism | N | Average | Standard Deviation | Standard Error | Minimum | Maximum |
|---|---|---|---|---|---|---|
| Santiago | 18 | 12.7369 | 1.6148 | 0.3806 | 10.4600 | 17.1200 |
| Coastal—no French Way | 18 | 10.7157 | 1.3041 | 0.3074 | 7.8000 | 12.2333 |
| Inland—no French Way | 18 | 15.6854 | 3.8943 | 0.9179 | 9.5000 | 22.9333 |
| Total | 54 | 13.0460 | 3.2387 | 0.4407 | 7.8000 | 22.9333 |

As there are more than two groups, we applied the one-factor ANOVA. According to Levene's test, at the 5% significance level, the null hypothesis of the equality of variances could be rejected. With respect to normality, at the 5% level, the null hypothesis of normality could be rejected in the case of the ACs through which the French Way of St. James passes. If we take into account the small values of the skewness and kurtosis statistics (1.3300 and 1.9534, respectively), it can be claimed that the deviation from normality is not relevant.

In view of the above, the Brown–Forsythe statistic had to be used instead of the standard F-statistic, concluding clearly (*p*-value < 0.0001) that the means differ from each other. To be more specific, we then carried out the Games–Howell post-hoc test, the results of which allow us to assert, for a significance level of 5%, that: (1) in the ACs crossed by the French Way of St. James, the importance of cultural tourism is, on average, significantly higher than in the coastal ACs through which it does not pass, the magnitude of the difference being close to 2%; (2) on the contrary, the ACs of the French Way of St. James show a lower level of cultural tourism than the inland ACs through which the French Way of St. James does not pass, with the difference being a bit higher than in the former case (about 3%); (3) in the ACs not crossed by the French Way of St. James, the inland regions have much higher levels of cultural tourism than the coastal regions (around 5% more).

Finally, we analysed within the group of ACs of the French Way of St. James whether there were relevant differences between the only one with coastline (Galicia) and the inland ACs (Aragón, Castilla y León, Navarra and La Rioja). According to the descriptive data (see Table 4), cultural tourism is higher in the coastal ACs than in the inland ACs. However, to verify this in a precise way, statistical techniques are needed.

**Table 4.** Descriptive statistics for the ACs of the French Way: coastal (Galicia) and inland.

| Cultural Tourism | N | Average | Standard Deviation | Standard Error | Minimum | Maximum |
|---|---|---|---|---|---|---|
| Coastal | 18 | 13.6167 | 1.8376 | 0.4331 | 10.1000 | 16.4000 |
| Inland | 18 | 12.5097 | 2.0155 | 0.4751 | 9.8000 | 17.6000 |
| Total | 36 | 13.0632 | 1.9820 | 0.3303 | 9.8000 | 17.6000 |

As in the first case, we can use a *t*-test for two independent samples. The Shapiro–Wilk test for normality showed that, at the 5% significance level, the null hypothesis of normality could not be rejected, both for the coastal AC and for the inland ACs. Levene's test revealed that the null hypothesis of homoscedasticity could also not be rejected. Consequently, we had to apply a test on the difference of means of two independent normal populations whose variances are unknown but can be assumed to be equal. This test showed that, at

the 5% level, we could state that cultural tourism is more relevant (slightly more than 1%) in the coastal AC.

*4.2. Websites*

The results are structured based on the parameters and variables analysed: positioning, engagement, audience origin and access devices.

4.2.1. Positioning

We analysed the positioning of the public websites of the official tourism promotion bodies of the Spanish ACs through which the French Way of St. James passes, both inland and coastal. This parameter is an indicator of the ratio of the websites at both the global and national level.

The world ranking (Figure 2), where the rank is defined by the highest sum of monthly unique visitors and page views in the world, shows for the confinement months of 2020 (March, April, and May) that Turismo de Galicia reaches the best position (252,825) of the CAs included, followed by Turismo de Castilla y León (301,919) and Turismo de Aragón (536,196). The rest of the promotional bodies of the Spanish communities along the French Way of S. James have a worse place: Turismo de La Rioja is in position 902,341 and Turismo de Navarra does not even enter the rankings due to its very low records, a constant repeated in the remaining elements of the positioning analysis, for both this year and 2021. In the analysis of the same months of the Holy Year 2021, a significant improvement can be seen in the rankings of Galicia and Aragón, rising to positions 135,117 and 393.958, respectively, which allows the Galician community to climb more than one hundred thousand positions in the global ranking, perhaps due to the interest aroused by the Holy Year and the opening of borders and the improvement in health on the road to the new normality. The opposite is the case of La Rioja and Castilla y León, which fall in the global ranking.

| **2020** | | **2021** | |
|---|---|---|---|
| turismo.gal | #252,825 | turismo.gal | #135,117 |
| lariojaturismo.com | #902,341 | turismo.navarra.es | - |
| turismo.navarra.es | - | turismodearagon.com | #393,958 |
| turismocastillayleon.com | #301,919 | lariojaturismo.com | #1,040,092 |
| turismodearagon.com | #536,196 | turismocastillayleon.com | #567,570 |

**Figure 2.** World ranking of official Spanish websites, 2020–2021.

If the parameter analysed is the national position, this is defined by the main website selected and indicates the ranking of the websites in the country itself. The rank (defined by the highest sum of monthly unique visitors and page views) reveals that the ranking for Turismo de Galicia in this case behaves similarly to the world ranking both in 2020 and 2021, where there are significant rises (Figure 3), being the first website of those analysed, with position 7016 in 2020 and 3038 in 2021, respectively (therefore, it has improved outstandingly). Behind, the rest of the ACs have suffered significant falls in 2021: Castilla y León or La Rioja, for example, from 8188 and 30,379, respectively, in 2020, to 16,332 and 32,115, respectively, at present.

According to the results obtained, during the confinement period, Turismo de Galicia, the website of the only coastal region, was the best positioned in relation to the rest of the inland regions analysed, in all the web rankings (world and national) and both in the months of forced confinement and at the beginning of the opening of travel, followed by Castilla y León and Aragón. At the other end of the scale is Turismo de Navarra, whose low website performance does not allow it to be in these rankings.

|  | 2020 |  |  | 2021 |  |
|---|---|---|---|---|---|
| 🏃 | turismo.gal | #7,016 | 🏃 | turismo.gal | #3,038 |
| LRT | lariojaturismo.com | #30,379 | 🔵 | turismo.navarra.es | - |
| 🔵 | turismo.navarra.es | - | A | turismodearagon.com | #10,024 |
| 🟫 | turismocastillayleon.com | #8,188 | LRT | lariojaturismo.com | #32,115 |
| 🔺 | turismodearagon.com | #15,112 | 🔵 | turismocastillayleon.com | #16,332 |

**Figure 3.** National ranking of official Spanish websites, 2020–2021.

4.2.2. Engagement

The second parameter is engagement, which provides information on the commitment between a website and the user visiting it and the level of interaction and degree of linkage between the user and the brand.

Adequate engagement is a desired objective for companies because it promotes customer loyalty so that committed consumers who identify with their destination/brand become its ambassadors, generating new customers thanks to their recommendations.

For its study, analysts consider a series of metrics that can help quantify the degree of user interaction with the destination/brand website, such as: total number of visits to the website, average duration of each session and bounce rate.

It should be remembered that the data analysed reflect the situation of the variables during the three months of home confinement forced by the pandemic (March, April and May 2020) and the same in 2021, the Xacobean Holy Year. The data are based on Figure 4, which shows the number of total visits or visits over time received by the websites of the communities through which the French Way of St. James passes in Spain during the months.

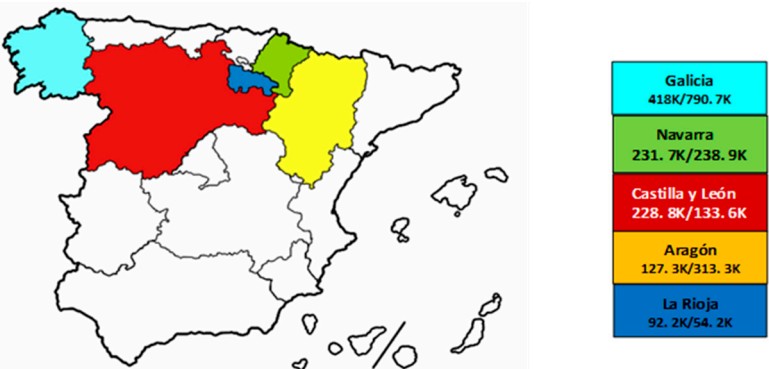

**Figure 4.** Visits to the websites over the months of analysis for the years 2020, 2021. Visits over time.

In 2020, Galicia's site once again stands out from the rest of the ACs with more than 400,000 visits, which means an average of 139,587 visits per month. At the opposite extreme is La Rioja, with just over 90,000 visits and 30,736 per month, the lowest figure of the cases analysed. Galicia also stands out because its visits are almost double those of Navarra and Castilla y León, reaching almost 420,000 compared to 231,000 and 228,000, respectively, whereas Aragón does not reach 130,000 visits and La Rioja reaches 93,000 visits, which means only 30,736 visits per month on average for the AC that is positioned at the bottom of the ranking.

The situation in 2021 does not differ too much in relation to the position of each community in the ranking; however, when analysing the number of visits, there are novelties as two portals stand out from the rest, Galician and Aragón (almost 200,000 more visits). The Galician coastal community has managed to practically double its visits compared to the same period last year, receiving 790,800 visits, which could be attributed to the increase in tourist movements favoured by the incipient opening of travel and coinciding with the Holy Year 2021. It seems that the taking advantage of the exceptional

nature of this year by the public bodies promoting tourism in Galicia has allowed them to undertake initiatives of universal significance that value the historical role of Galicia as the epicentre of the French Way of St. James, generating a significant increase in the number of visits.

Another of the variables to be analysed in relation to engagement is the average duration of visits to each of the websites of the ACs under study. It represents the average time that the user spends on the website, and it is important that it is as high as possible, since this means that the website, its structure and content are interesting for users, and will help with the organic, natural or SEO (Search Engine Optimisation: a set of actions aimed at improving the organic or natural positioning of a website in the list of results from Google or other internet search engines) positioning of the pages.

In Figure 5, regarding the duration of users' visits to websites in 2020, the most striking datum corresponds to Turismo de La Rioja, with 13 min, despite being an AC with a low positioning in general and with few visits in particular, which shows the interest of its page, whose content and structure attract a great deal of user attention. Castilla y León and Aragón are also two communities that stand out in this variable with 11 and 8 min respectively, maintaining the relationship with positioning. Galicia and Navarra appear in the last place, with only 2 min of permanence. It is important to highlight how Galicia, being the best positioned website and the one with the highest number of visits (Figures 4 and 5), does not translate this relevance into the time that its users spend on its portal, which could mean that the content, structure and/or usability in 2020 were not sufficiently attractive to users.

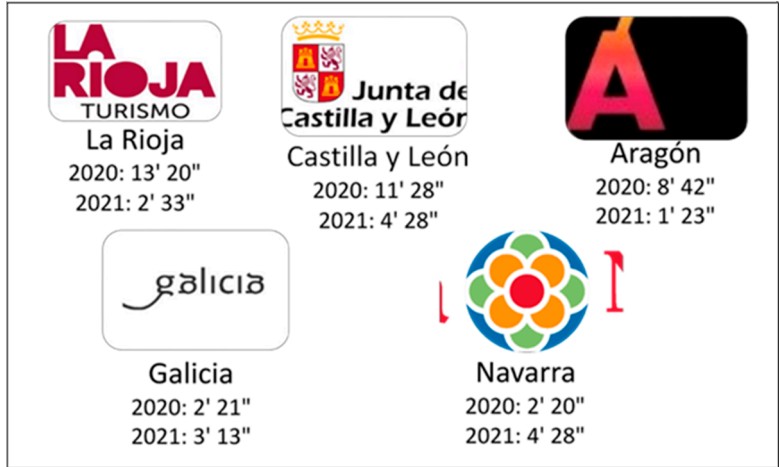

**Figure 5.** Average duration of visits to the websites analysed.

In 2021, the situation has changed as the time spent on the sites of La Rioja, Castilla y León and Aragón has diminished in a worrying way, but has increased in Galicia and Navarra, making them, together with Castilla y León, the communities with the longest permanence. These destinations seem to have managed to capture visitors' attention and have adapted to what interests them, in addition to possible improvements in the usability and structure of their sites.

To finish with the analysis of engagement, we must also refer to the bounce rate, which measures the number of sessions in which the user has visited a single page of the site and has left without interacting with it.

A high bounce rate means that the time spent by visitors on a page is very short (only a few seconds) and that they do not browse other subpages of the website. Conversely, the lower the bounce rate, the longer users stay on the same page and the longer they navigate through different subpages or sections of the site. We must be aware that Google translates this data into quality, so if the bounce rate is high, it will consider that the user has not

found relevant content and will penalize the site; conversely, if it is low, it implies that it is a useful website for visitors and will give it greater visibility (i.e., better position) [33].

However, a very high rate is not always alarming, as this measure should be studied together with the dwell time, which is the time that elapses from when the user clicks on the search result until he/she leaves the website. The user may have a specific need, spend time on the page reading the result and interacting and then leave, which would cause a high dwell time and lead search engines such as Google to recognize permanence as something positive and improve the SEO positioning of the page, even if the bounce rate is high [34].

According to Figure 6, most of the websites analysed have a bounce rate between 50% and 70%. The positive results of Castilla y León in 2020 stand out, with a bounce rate of 50.84% and an average of 9 pages visited with a duration of 11 min and 28 s per user, which shows the interest aroused by its portal apart from reaching a better positioning. La Rioja also are remarkable, with a bounce rate of 58.65%, 3.95 pages viewed and the longest duration of the visits, 13 min and 20 s, respectively. However, in 2021, the values have worsened slightly in both cases, with bounce rates exceeding 60% and a decrease in the number of page views.

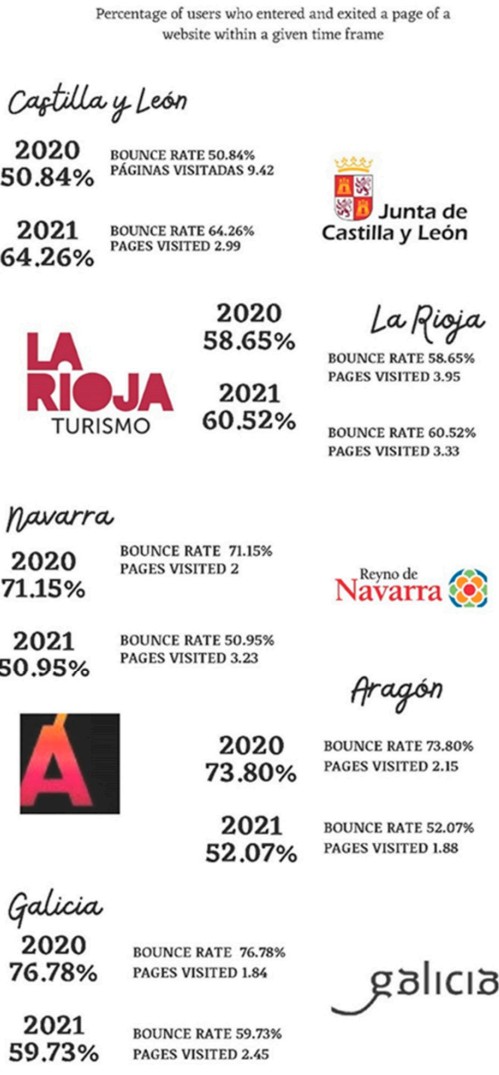

**Figure 6.** Bounce rates March/May 2020–2021.

In 2020, the bounce rates for Navarra, Aragón and Galicia were a little high, exceeding 70%. Possibly these data are related to the variable 'page views' as none of them manage to get users to see more than two pages and, on the other hand, the 'visit duration', especially in the cases of Navarra and Galicia, is very low (about 2 min). However, these same communities have improved their metrics in 2021 by reducing the bounce rate between 50% and 60% and increasing the number of page views and the duration of visits.

According to Merodio [33], the value of an acceptable bounce rate depends, apart from the quality of the content of the destination page, on the amount of traffic it receives and the type of website. However, there are some standard values: a bounce rate of less than 50% would be excellent, making business revenue increase considerably and the position achieved by the site in Google could be among the highest, while bound rates above 70% would not be of any benefit to the website positioning.

### 4.2.3. Origin of the Audience

The origin of the users of the portals of the ACs analysed is another of the parameters that provide insight into the functioning of their websites. According to the data collected, 84.99% in 2020 and 88.79% in 2021 of the web users of the portals of the tourist promotion institutions of ACs are from Spain, whereas only 15.01% in 2020 and 11.21% in 2021 are of international nationality (France, Italy, the United States, Germany, Portugal, Argentina, the United Kingdom, Belgium and Poland). The Figure 7 shows the countries that generate the most traffic for the websites analysed, accounting for more than 90% of the total in both years.

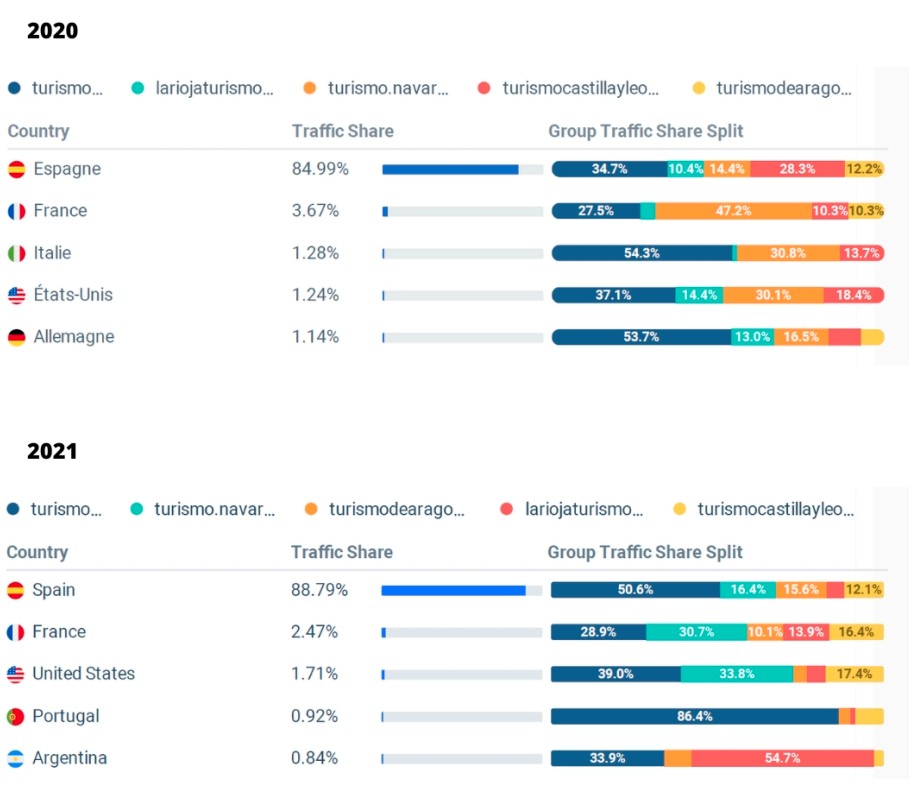

**Figure 7.** Origin of users of the sites of the ACs analyzed 2020–2021.

Concerning the data obtained through the Similar Web tool, of the national visitors to the websites in both years, Galicia receives the highest number of users (34.66% and 50.6%), corroborating the better positioning found in the previous analysis. If we look at the origin of foreign visitors, in 2020 the French led the visits to the Navarra website (47.22%), possibly due to territorial proximity, as did the Portuguese, who in 2021 were the main visitors to the Galicia website (86.4%). In addition, in both years, the Galician site is

the most often visited by Italian, German and American people, which again corroborates the data for the better positioning of this portal. This is logical, since it is the AC with a more direct link to the French Way of St. James as it is the final destination of the itinerary where Apostle St. James' remains are located.

### 4.2.4. Access Devices

To finish the study of the performance of the web portals of the Spanish ACs crossed by the French Way of St. James, we analysed data on the devices employed by users to access the websites (Figure 8). The general trend of content consumption through mobile devices, in 2020 and 2021, also bursts with force in the tourist ecosystem, as corroborated by users who access the websites of Galicia and Navarra in the first year of analysis and Galicia and Aragon in the second. There are exceptions, such as La Rioja and Castilla y León, which have the highest percentage of access via computer. Although these are general data and must be contextualised with the data on internet connection penetration and use of mobile technologies in each country, they serve to perceive a general trend of increasing content consumption from mobile devices at a worldwide level.

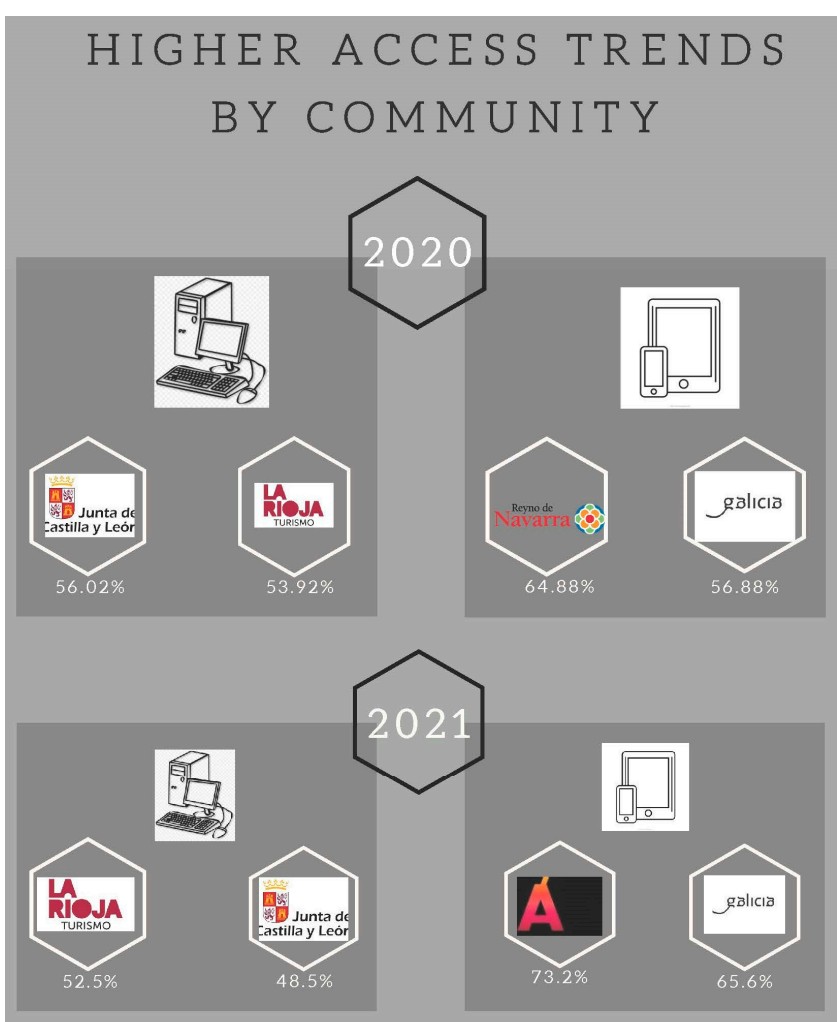

**Figure 8.** Access devices to the analyzed web portals 2020–2021.

### 5. Conclusions

The advent of the post-pandemic will affect virtually every aspect of travel, and most destinations, including the cultural ones. Travellers' preferences will continue to evolve, so the travel industry, both in Spain and at global level, will need to be agile to meet the new demands of tourists. Perhaps the COVID-19 pandemic will be the genesis of change and

will serve as a transformational lever to help relaunch an impoverished tourist industry with an obsolete model in need of change, generating an additive process between the public and private spheres.

While no one in the industry can have full control over the recovery, we are witnessing promising signs as borders reopen, especially in some European countries, although demand stimulation mechanisms will be needed when, through vaccines, the pandemic is brought under control and thus the economy is invigorated. Now is the time to sow in order to start harvesting when this cyclone has passed.

In this context, cultural tourism can be an important locomotive for the recovery of a tourist sector that wants to re-emerge as a response to a very uncertain pandemic situation, and the French Way of St. James in Spain can offer tourists what they are asking for: a sustainable, safe product adapted to the new consumer's needs.

The twofold methodology employed in this research, based on the analysis of data from the Ministry of Culture [28] and the online tool Similar Web, has allowed us to study cultural tourism and the websites of the different Spanish communities along the French Way, which has led to a series of conclusions that we present below.

Firstly, from the Ministry data, without considering the French Way of St. James as a factor, at a significance level of 5%, we can affirm that cultural tourism is more relevant in the inland ACs. One possible interpretation of this result is that in these ACs, the absence of the alternative for sun and beach tourism implies that individuals consume more cultural tourism, whereas, on the contrary, in the coastal ACs, sun and beach tourism competes with the consumption of cultural tourism, causing it to be reduced.

If we incorporate the French Way of St. James into the analysis, for the same level of significance, we conclude that: (i) in the ACs crossed by the French Way of St. James, the importance of cultural tourism is, on average, significantly higher (difference close to 2%) than in the coastal ACs out of that route; (ii) on the contrary, the ACs along the French Way of St. James have a lower level of cultural tourism than the inland ACs out of it, with the magnitude of the difference being somewhat greater than in the previous case (around 3%); and (iii) in the ACs not crossed by the French Way of St. James, the inland ACs have much higher levels of cultural tourism than the coastal ACs (about 5% more).

In view of these results, it seems that, with regard to cultural tourism, on the one hand, the determining factor is geographical, i.e., whether or not there is a coastline; specifically, the existence of a coastline acts as a negative factor for this type of tourism. On the other hand, the French Way of St. James does not appear to act as a driving force for cultural tourism; quite the reverse, it seems that those who follow the French Way of St. James would rather pursue other types of interests (scenic, spiritual, etc.) than those of a cultural nature.

Secondly, the analysis of the data extracted with the Similar Web tool from the tourist promotion portals of the destinations analysed led to the following conclusions.

Throughout the period studied, the pages analysed have experienced a notable growth in their use, highlighting their relevance for tourist organisations and destinations as a promotional and marketing tool.

Galicia's promotional website stands out in this analysis. It is the AC that best uses its online platform as a means of dissemination, content generation and communication that favours the exchange of information with its users and community image. In the parameters of positioning with other websites, it is in a higher position in relation to the rest of the ACs, both in the world and national rankings. In addition, it appears as the destination with the highest engagement. However, the bounce rate should be improved.

On the other hand, the parameters of origin show that since the beginning of the pandemic, domestic tourism has become more important in the destinations analysed to the detriment of international tourism. This corroborates the data on tourist movements from Frontur [2] surveys, which show that tourist activity is currently recovering in Spain thanks to domestic movements.

Finally, the devices used for accessing information and the process of digital convergence in which we are immersed have resulted in an increasingly greater and more versatile use of smartphones, which favours access to tourist promotion websites, an aspect that these organisations must take into account in order to optimise their sites.

This research shows that cultural tourism is a more powerful element of attraction for the coastal ACs analysed than for the inland communities, which changes the paradigm established until now, which stated that the coastal communities had their sun and beaches as the main focus of attraction. However, it still may be necessary to identify the keys that allow for understanding the potential of cultural tourism to contribute to overcoming the crisis, as well as its response to the demands of new post-pandemic tourist behaviour.

The tourism potential that the French Way of St. James could potentially generate is, by all accounts, largely untapped. While it is true that public administrations have made a considerable effort to promote tourism promotion websites, this does not seem to be significant in order to improve the role of the French Way of St. James as an additional factor of tourist attraction. It is therefore necessary that public authorities explore other options to try to adequately exploit the tourism potential of the French Way of St. James.

**Author Contributions:** C.R.-V.: Software, Investigation, Writing—Original Draft, Methodology, Visualization, Supervision; P.C.-G.: Formal analysis, Investigation, Writing—Original Draft, Methodology, Visualization, Supervision; V.A.M.-F.: Conceptualization, Writing—Review & Editing, Project administration. All authors have read and agreed to the published version of the manuscript.

**Funding:** This research received no external funding.

**Institutional Review Board Statement:** The study did not require ethical approval.

**Informed Consent Statement:** Not applicable.

**Data Availability Statement:** Data supporting reported results can be found https://www.culturay deporte.gob.es/servicios-al-ciudadano/estadisticas/cultura/mc/naec/portada.html.

**Conflicts of Interest:** The authors declare no conflict of interest.

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
