# Peer review of "Cultural Tourism in a Post-COVID-19 Scenario: The French Way of Saint James in Spain from the Perspective of Promotional Communication"

_societies, doi:10.3390/soc13010016_

Round 1
Reviewer 1 Report
Dear Author,
The article titled "Cultural Tourism in the Postcoital Scenario: The French Way of Saint James from the Perspective of Promotional Communication" aims to establish the importance of Cultural Tourism as a key travel motivation for Spanish residents in ACs through which the French Way of Saint James passes, and how they behave on the institutional websites of tourism promotion organizations.
The paper was well prepared, but it needs minor corrections.
After reviewing the paper, I have comments and suggestions to improve the paper as follows:
Titel
The title of the paper should be modified. It lacks a reference to the research area. I suggest supplementing the French Way of St. James in Spain in the title.
1.In the introduction
Only one issue of tourism and its changes over several decades was presented. Attention was paid to the crisis in tourism caused by Covid-19. However, information related to the topic of the article, i.e. cultural tourism, the French Way of St. James, was missing.
No information was presented: What is the current state of research on this topic? Why did the authors take up this topic?
At the end of this chapter, the purpose of the work should be stated.
The work does not pose specific research questions.
2. Theoretical basis and development of hypotheses
The theoretical part is well presented.
3. In Materials and methods
Please describe the research area in detail.
I propose to introduce the scheme of the research procedure.
Please describe in detail the research method used.
4. Results
The results are presented and described in a good way, they are very interesting and important for the development of Cultural Tourism. However, I would suggest doing some organizing of the obtained results in the form of tables. The text must be clear and easy to read.
5. The discussion part was missing,
The author should discuss and explain more the conclusions and results of the work. It is also important to describe the results of the work in more detail in this section. The author should compare his project and results with the results of similar ongoing research on this topic from other parts of Europe and around the world.
This section should still answer the question: what tangible benefits this study has brought to the development of cultural tourism and make recommendations. Emphasize which elements of cultural heritage are most important.
Technical errors to be corrected:
[387] - Please improve the quality of Figure 2 in accordance with the requirements of the journal Societes
[404] - Please improve the quality of Figure 3 in accordance with the requirements of the journal Societes
- [526] Please improve the quality of Figure 7 in accordance with the requirements of the journal Societes
Correct literature according to Journal.
All in all, I recommend this paper for publication in the Journal “Societes” after minor changes.
Kind regards,
Reviewer 2 Report
Comments for the paper
societies-2100220
Cultural Tourism in a Post Covid Scenario: The French Way of Saint James from the Perspective of Promotional Communication
Tourism has been one of the sectors most affected by the Covid-19 pandemic. One of the side effects of the pandemic is the demand for safe and quiet spaces, giving rise to the search for a new lifestyle, "slow living", which could represent an opportunity for Cultural Tourism. In this context, the main objective of this article is twofold: (i) to establish the relevance of cultural tourism for residents in Spain for the autonomous communities along the French Way of Saint James, and (ii) to determine their behaviour on their institutional tourism promotion websites. For our analysis, we use equality of means tests and ANOVA (for data from 2002-2020), as well as measures of positioning, engagement, origin of the audience and access devices (for data from 2020-2021). The main conclusion is that the Way of St. James does not act as a driving force for cultural tourism, even though the websites of tourism promotion organisations have experienced a remarkable growth in their use. The article develops an original relation of Cultural Tourism through the analysis of the French Way of St. James in Spain and the web positioning of official tourism promotion organisations before and during Covid-19.
My suggestions to the authors for the improvement of the paper relate to the following:
1. This paper is well-developed and interesting.
1. The authors should clarify if and how their approach differs from the previous literature in the Introduction of the paper. The paper's contribution to the international bibliography is not clearly mentioned in its current version of paper.
2. The authors enrich their references and analyze the determinants variables of tourism (Environmental Sustainability: Szolnoki and Tafel, 2022, health quality: Konstantakopoulou, 2022). These references are essential.
3. The authors should put more effort and thoroughly discuss point estimates, estimated effects, and the intuition behind the results backed up by the literature. Generally, the empirical part is poor; it is not documented why they choose the specific method they apply, etc.
References
Konstantakopoulou, I., 2022. Does health quality affect tourism? Evidence from system GMM estimates. Economic Analysis and Policy, 73, 425-440.
Szolnoki, G.; Tafel, M., 2022. Environmental Sustainability and Tourism—The Importance of Organic Wine Production for Wine Tourism in Germany. Sustainability, 14, 11831. https://doi.org/10.3390/su141911831
